# Exploring perspectives on artificial intelligence: awareness, attitudes, and knowledge among health majors students at Saudi universities



Mai Albaik[1], Saeed A. Al-Qahtani[2], Mohammad Jaffar Sadiq Mantargi[3], Adel Alghamdi[4], Ikhlas A. Sindi[5], Ryan A. Sheikh[6,7,8], Mohamed Kamel[9] and Lina A. F. Kurdi[10]

[1] Department of Chemistry, Preparatory Year Program, Batterjee Medical College, Jeddah, Saudi Arabia
[2] Department of Clinical Pharmacy, College of Pharmacy, Jazan University, Jazan, Saudi Arabia
[3] Department of Pharmaceutical Sciences, Pharmacy Program, Batterjee Medical College, Jeddah, Saudi Arabia
[4] Department of Biology, Preparatory Year Program, Batterjee Medical College, Jeddah, Saudi Arabia
[5] Department of Biology, Faculty of Sciences, King Abdulaziz University, Jeddah, Saudi Arabia
[6] Department of Biochemistry, Faculty of Science, King Abdulaziz University, Jeddah, Saudi Arabia
[7] Experimental Biochemistry Unit, King Fahd Medical Research Center, King Abdulaziz University, Jeddah, Saudi Arabia
[8] Cancer and Mutagenesis Unit, King Fahd Medical Research Center, King Abdulaziz University, Jeddah, Saudi Arabia
[9] Department of Medicine and Infectious Diseases, Faculty of Veterinary Medicine, Cairo University, Cairo, Egypt
[10] Department of Biological Sciences, College of Science, University of Jeddah, Jeddah, Saudi Arabia

Corresponding author
Mai Albaik, mai.albaik@bmc.edu.sa

## ABSTRACT

**Background:** The world is witnessing tremendous development in the field of new digital tools, including artificial intelligence (AI), in all sectors, including the health and medical sectors. However, educational and training opportunities in the field of artificial intelligence remain nascent and limited. Hence, this study aims to assess the awareness, attitudes, and knowledge of artificial intelligence among students of health specialties in Saudi universities and to assess whether artificial intelligence is viewed as a beneficial innovation or a potential threat to healthcare roles.

**Methods:** This cross-sectional study included 498 male and female students from various health colleges at different Saudi universities. The participants completed an online questionnaire adapted from previous studies to assess their awareness, attitudes, and knowledge of artificial intelligence. Descriptive statistics and chi-square analyses were conducted to explore the associations between variables related to artificial intelligence and other factors.

**Results:** Most students showed a high level of awareness of artificial intelligence, with social media being identified as their main source of information about artificial intelligence. While students' attitudes towards AI were generally positive, for example, 89.2% of the students believed that AI would be crucial to the future of healthcare, 76.7% supported AI education, and 78.3% were keen to increase their knowledge of AI. In terms of assessing students' knowledge of AI, the study revealed

that the participating students had moderate knowledge of AI principles and skills, with significant gaps in understanding specific AI capabilities and functions. **Conclusions:** While healthcare students in Saudi Arabia demonstrate strong awareness and positive attitudes towards AI, there are significant gaps in practical knowledge. These findings underscore the need for tailored educational strategies to better integrate AI into curricula, thus preparing future healthcare professionals to effectively leverage AI.

# INTRODUCTION

Artificial intelligence (AI) has become one of the most transformative innovations of the past decade, reshaping numerous sectors, including healthcare (*Vinothkumar & Karunamurthy, 2023*). AI is defined as the simulation of human perception and behaviour through the development of computers and technology to perform human tasks, such as visual perception, speech recognition, and decision-making (*Al Saad et al., 2022*). In healthcare, AI has been applied across diverse disciplines, including dermatology (*Hogarty et al., 2020*), ophthalmology (*Ting et al., 2019*), and pathology (*Teng et al., 2022*), supporting diagnostic accuracy, enhancing treatment planning, and improving patient care. Despite these benefits, concerns remain regarding reliability, ethical implications, and the potential displacement of human roles in healthcare (*Akgun & Greenhow, 2022*; *Lee & Yoon, 2021*).

This dual nature of opportunity and challenge underscores the need for AI education in healthcare. Globally, there is growing consensus that healthcare students must be equipped with the necessary knowledge and skills to engage effectively with AI-driven tools (*Cheng et al., 2020*). Effective AI education should extend beyond theoretical instruction to include practical training, enabling students to gain hands-on experience with emerging technologies. However, the degree to which AI has been integrated into medical curricula varies considerably across regions and disciplines, with Saudi universities showing limited adoption and inconsistent coverage.

Recent studies have highlighted both enthusiasm and apprehension regarding AI integration in healthcare education. For example, *Veras et al. (2024)* conducted a mixed-methods crossover randomized controlled trial in which health sciences students evaluated ChatGPT-3.5 for usability. Findings revealed perceived benefits, including productivity, creativity, and enhanced learning support, alongside concerns about accuracy, academic integrity, and the lack of clear institutional guidelines. These insights emphasize the importance of exploring students' awareness, attitudes, and knowledge regarding AI to inform evidence-based curricular development (*Veras et al., 2024*).

Research highlighting the importance of AI perceptions in healthcare disciplines has proliferated worldwide (*Diaz et al., 2021*; *Sit et al., 2020*), in the Middle East (*Al Saad et al.,*

*2022*; *Hassan Mekawy, Ali Mohamed Ismail & Zayed Mohamed, 2020*) and in Saudi Arabia (*AlAhmari, 2022*; *Khanagar et al., 2021*).

Various studies with limited focus on their specific disciplines like medicine and dentistry, have provided valuable insights, the most concerning factors are small sample size and single institutional studies. Moreover, these studies have not examined the student's perceptions across multiple health related majors or geographic regions. There is a lack of multi-centered, statistically stable sampled sized studies with holistic approach of how healthcare students across the Saudi Arabia perceive AI. The current study addresses this gap by investigating awareness, attitude and knowledge among students from various health majors across 11 universities of Saudi Arabia. By this study, a novel, generalised findings that can inform AI focused curriculum design, policy formulation and workforce development strategies personalised to the Saudi healthcare systems can be addressed.

However, a study intended to study the dental faculty's awareness and attitudes toward AI in dentistry using a mixed-method approach 74% participants expressed having an idea regarding the basic AI applications and a few 16.5% expressed familiarities with advanced versions of AI applications including machine learning and deep learning. The study recommends AI training workshops, curriculum updates, and policy changes to bridge knowledge gaps. It highlights the need to align dental education with AI advancements for future-ready practitioners (*Abdullah et al., 2025*).

A systematic review comprising of 253 studies identified five key points for developing a responsible AI *i.e.*, sustainability, human-centeredness, inclusivity, fairness, and transparency. The ethical bias such as algorithm bias, data privacy, deployment of AI in low-income settings. It has proposed a guide for policymakers, healthcare providers and AI developers for proper application. The review underscores the importance of interdisciplinary collaboration and robust governance to ensure AI benefits all stakeholders (*Ali Salem Maharem, 2024*).

One such similar study published identified existing research gaps in the understanding of AI, such as limited exposure to AI training, geographical disparities, lack of AI related skill development, longitudinal and comparative data, ethical and social implications, lack of interdisciplinary collaborations and lack of assessment tools (*Mousavi Baigi et al., 2023*; *Serbaya et al., 2024*). Similarly, another study identifies lack of AI topics in medical education and considered as barrier. The students have to be better prepared for the integration of AI in the curriculum; however, the students are interested but lack complete idea regarding the utilisation of AI (*Al-Qerem et al., 2023*; *Mehta et al., 2021*). Another study reflecting the Jordanian approach recommends developing user friendly AI tools, big-data analytical tools, providing training for healthcare professional, as the gap existing in the current literature, which perfectly aligns with the current study (*Al-Dmour et al., 2025*).

Therefore, this study aims to assess awareness, attitudes, and knowledge regarding AI among health sciences students across 12 Saudi universities, while examining whether AI is perceived as a beneficial innovation or a potential threat to healthcare careers. Findings are expected to support curriculum enhancement, policy formulation, and workforce development strategies tailored to the Saudi healthcare context.

## SUBJECTS AND METHODS

### Study type, sampling technique, sample size and participants

This study adopted a cross-sectional survey design, in accordance with the Strengthening the Reporting of Observational Studies in Epidemiology (STROBE) and Survey Reporting Guideline (SURGE) recommendations. Stratified sampling was employed to ensure representation across diverse academic levels (preparatory to postgraduate) and institutional types (governmental and private). Participants were health sciences students aged 21–45 years, recruited from 12 Saudi universities, including Batterjee Medical College (Jeddah), King Abdulaziz University (Jeddah), Jeddah University (Jeddah), Umm Al Qura University (Makkah), King Saud University (Riyadh), King Faisal University (Riyadh), King Khalid University (Abha), King Fahd University of Petroleum and Minerals (Dhahran), Taibah University (Madinah), AlRayan College (Madinah), Jazan University (Jazan), and Tabuk University (Tabuk).

#### Sample size

The minimum sample size was calculated *via* Raosoft® (http://www.raosoft.com/samplesize.html), with parameters set at a 95% confidence level, a margin of error not exceeding 5%, and an expected prevalence of 50%, resulting in a required sample size of 377 students. A total of 498 students completed the survey, exceeding the calculated sample size and ensuring robust statistical power.

### Questionnaire design and data collection

The questionnaire utilized in this study was designed to assess awareness, attitudes, and knowledge toward artificial intelligence among health majors' students in Saudi universities. Students were asked to complete an online questionnaire adapted from previous studies (*Al Saad et al., 2022*; *Hassan Mekawy, Ali Mohamed Ismail & Zayed Mohamed, 2020*; *Sit et al., 2020*). The study utilized a questionnaire as its primary tool, which included an introductory section providing background information about the researchers and the objectives of the study, as well as a section for obtaining participant consent (*Easwaran et al., 2024*).

The questionnaire consisted of 22 questions organized into four sections. The first section comprised seven demographic questions addressing gender, age, nationality, university, major, academic year, and current grade point average (GPA). The second section assessed awareness of AI through questions on its definition and sources of information (*e.g.*, academic courses, social media). The third section evaluated attitudes toward AI, exploring perceptions of its role in healthcare, its potential impact on medical careers, and the necessity of integrating AI training into curricula. The fourth section measured knowledge of AI, including understanding of core principles, types of AI (*e.g.*, Narrow, General, Super AI), and cognitive skills related to AI programming, such as learning and reasoning.

The response options were categorized as "agree," "disagree," or "do not know." A scoring system was implemented by assigning a value of "1" for each correct response and "0" for incorrect or unsure responses.

To ensure validity, the questionnaire was reviewed by experts in AI and medical education, and a pilot test was conducted. The pilot test involved 15 students (not included in the main study) to assess clarity, question flow, and estimated completion time. The average completion time was 7–9 min. Feedback led to minor revisions, including rewording two questions for clarity, adding brief explanations of AI terminology, and improving response options to reduce ambiguity. Internal consistency was confirmed using Cronbach's alpha, yielding satisfactory reliability coefficients for awareness ($\alpha$ = 0.78), attitude ($\alpha$ = 0.81), and knowledge ($\alpha$ = 0.76).

The questionnaire was bilingual (English-Arabic) to accommodate participants, with the validity of the translated version confirmed through forward and backward translation.

### Informed consent and ethical consideration

Ethical approval for the study was obtained from the Institutional Review Board of Batterjee Medical College, Jeddah (Reference No. RES-2023-0055). The written informed consent was obtained from all participants prior to their involvement in the study. For the online survey, participants were required to read and electronically approve the consent form before completing the questionnaire. Additionally, written and signed ethical consent forms were collected from participants as part of the ethical requirements.

### Statistical analysis

The survey data were initially entered into a Microsoft Excel spreadsheet, coded appropriately, and saved as a CSV file. This file was then imported into IBM SPSS Statistics version 26 (IBM Corp., Armonk, NY, USA) for analysis. The questionnaire was structured into four main sections: (1) demographic characteristics, (2) awareness of artificial intelligence (AI), (3) attitudes toward AI, and (4) knowledge of AI. Descriptive statistics, including frequencies and percentages, were used to summarize responses for all variables and are presented in Table 1.

The data was coded aligning with practices in survey research *i.e.*, the binary questions are coded as 1 = yes and 2 = no; if third option is included *i.e.*, 3 = do not know. Coding for the source of information was done as 1 = academic courses, 2 = print media (books, magazines, newspapers), 3 = broadcast media (TV, radio), 4 = outdoor media (*e.g.*, billboards, transit ads, …), 5 = social media (Facebook, Instagram, Snapchat, Twitter,…), 6 = Internet (web pages, online databases, …), 7 = video games, 8 = family members, 9 = friends and 10 = other and following question and other options in the similar manner. Chi-square ($\chi^2$) tests were applied to examine associations between key categorical demographic variables (such as gender, academic year, type of university, and GPA category) and outcome variables, including levels of AI awareness (yes/no), attitudes (*e.g.*, support for AI integration in education, willingness to learn about AI), and knowledge categories (low, moderate, high). Pearson's chi-square test was used to identify statistically significant relationships, and the results were interpreted at three significance thresholds: $p \leq 0.05$ (significant), $p \leq 0.01$ (highly significant), and $p \leq 0.001$ (very highly significant). These analyses aimed to highlight variations in AI-related perceptions and understanding across different student subgroups. Before performing the chi-square analysis, all categorical variables were tested

**Table 1 Demographic details of the study participants.**

| Variables | Number (Percentage) |
|---|---|
| Gender | |
| Male | 328 (65.9) |
| Female | 170 (34.1) |
| Age | |
| 15–20 years | 200 (40.2) |
| 21–25 years | 258 (51.8) |
| 26–30 years | 16 (3.2) |
| 31–35 years | 10 (2.0) |
| 36–40 years | 10 (2.0) |
| 41–45 years | 4 (0.8) |
| Nationality | |
| Saudi | 452 (90.8) |
| Non-Saudi | 46 (9.2) |
| University | |
| Public universities | 370 (74.1) |
| Private universities | 128 (25.7) |
| Program | |
| Medicine | 236 (47.4) |
| Pharmacy | 62 (12.4) |
| Dentistry | 18 (3.6) |
| Nursing | 24 (4.8) |
| Radiology | 10 (2.0) |
| Physical therapy | 2 (0.4) |
| Respiratory therapy | 20 (4.0) |
| Occupational therapy | 12 (2.4) |
| Healthcare administration | 6 (1.2) |
| Clinical nutrition | 14 (2.8) |
| Medicinal laboratories | 46 (9.2) |
| Preparatory year | 42 (8.4) |
| Year of study | |
| Preparatory year | 50 (10.0) |
| Preclinical year | 274 (55.1) |
| Clinical year | 132 (26.5) |
| MSc or PhD | 10 (4.0) |
| GPA | |
| ≤3.5 | 66 (13.3) |
| 3.5–4 | 104 (20.9) |
| 4–4.5 | 154 (30.9) |
| >4.5 | 174 (34.9) |

for compliance with the statistical norms. For contingency table we verified that expected cell frequencies exceeded five meeting the fundamental requirement for chi-square test. This method ensured the validity of all tests of independence between categorical variables while maintaining statistical strength.

## RESULTS

### Descriptives of the participants

The investigation involved a total of 498 individuals whose demographic information is outlined in Table 1. The majority of the sample consisted of males, encompassing 328 individuals (65.9%), and 170 were females (34.1%). Analysis of the age distribution revealed that most participants belonged to the 21–25 age bracket (258 participants, 51.8%), followed by the 15–20 age bracket (200 participants, 40.2%). A smaller percentage of participants fell into the 26–30 years age groups (16 participants, 3.2%), 31–35 years (10 participants, 2.0%), 36–40 years (10 participants, 2.0%), and 41–45 years (four participants, 0.8%). The majority of participants were identified as Saudi nationals (452 participants, 90.8%), whereas individuals of non-Saudi origin constituted a minority (46 participants, 9.2%). The participants primarily hailed from public academic institutions (370 participants, 74.1%), in contrast to those from private universities (128 participants, 25.7%).

The academic disciplines that were present in the study included various fields, such as medicine (236 participants, 47.4%), pharmacy (62 participants, 12.4%), dentistry (18 participants, 3.6%), nursing (24 participants, 4.8%), radiology (10 participants, 2.0%), physical therapy (two participants, 0.4%), respiratory therapy (20 participants, 4.0%), occupational therapy (12 participants, 2.4%), healthcare administration (six participants, 1.2%), clinical nutrition (14 participants, 2.8%), medicinal laboratories (46 participants, 9.2%), and preparatory years (42 participants, 8.4%).

In terms of academic progress, the largest proportion of participants were enrolled in the preclinical year (274 participants, 55.1%), followed by the clinical year (132 participants, 26.5%), the preparatory year (50 participants, 10.0%), and the MSc or PhD level (10 participants, 4.0%). This distribution indicates that the majority of respondents were still in the early or mid-stages of their academic training. As such, their perspectives on AI may reflect foundational exposure rather than clinical experience or postgraduate specialization. Understanding their level of academic progress is important, as it may influence their awareness and attitudes toward AI depending on whether they have encountered AI applications during clinical training or coursework.

Analysis of the distribution of GPAs revealed that 34.9% of the participants had a GPA exceeding 4.5, 30.9% held a GPA ranging between 4 and 4.5, 20.9% attained a GPA between 3.5 and 4, and 13.3% obtained a GPA of 3.5 or lower.

### Awareness of AI

The level of AI awareness among the participants was assessed through a series of questions, and the outcomes are delineated in Table 2.

**Table 2 Chi-square analysis of awareness against the categorical variables.**

| Demographic/ Continuous variable | Awareness (Categorical variable) | $\chi^2$ value | Degree of freedom | Statistical significance | Significance level |
|---|---|---|---|---|---|
| Gender | What is the definition of artificial intelligence (AI)? | 14.205 | 2 | 0.001 | Highly significant |
| | Did you attend courses about artificial intelligence and data science in the last 5 years?—No | 28.575 | 2 | 0.000 | Extremely significant |
| Age | Did you attend courses about artificial intelligence and data science in the last 5 years?—No | 62.518 | 42 | 0.022 | Significant |
| Nationality | What are your sources of information about artificial intelligence?—Academic courses | 16.372 | 6 | 0.012 | Significant |
| University | Did you attend courses about artificial intelligence and data science in the last 5 years? | 17.741 | 2 | 0.000 | Extremely significant |
| Program | What is the definition of Artificial Intelligence (AI)? | 36.398 | 22 | 0.028 | Significant |
| | Did you attend courses about artificial intelligence and data science in the last 5 years? | 131.085 | 22 | 0.000 | Extremely significant |
| Year of study | Did you attend courses about artificial intelligence and data science in the last 5 years? | 80.893 | 16 | 0.000 | Extremely significant |
| | What are your sources of information about artificial intelligence? | 30.293 | 18 | 0.035 | Significant |

### Definition of AI

AI was recognized by a large majority (96%) of the individuals surveyed. Gender was found to have a statistically significant relationship with this level of awareness ($\chi^2$ = 14.205, $p$ = 0.001). Additionally, programme and study year were also significantly associated ($\chi^2$ = 36.398, $p$ = 0.028; $\chi^2$ = 24.259, $p$ = 0.084).

### Attendance of AI and data science courses

Only 28.5% of the respondents had participated in courses related to AI and data science within the previous five years. The statistical analysis showed significant relationship between various variables of the study *i.e.*, gender showed a strong association with attendance of AI course ($\chi^2$ = 28.57, $*p*$ < 0.001), similar pattern of association was seen with the type of university ($\chi^2$ = 17.741, $*p*$ < 0.001) and study year ($\chi^2$ = 80.893, $*p*$ < 0.001) indicating the importance of AI and data science. In case of programs included in the study showed significance association ($\chi^2$ = 36.398, $*p*$ < 0.05) with the awareness related to AI. These findings underscore the importance of targeted educational strategies to address challenges associated with AI for better acceptability.

### Sources of information about AI

Participants reported diverse sources of information about AI (Fig. 1). Social media was the most common, followed by the internet, broadcast media, academic courses, family and friends, print media, video games, and outdoor advertisements.

### Attitude of AI

The level of AI attitude among health major students is presented in Table 3. A substantial majority of respondents (89.2%) perceived the anticipated role of artificial intelligence in

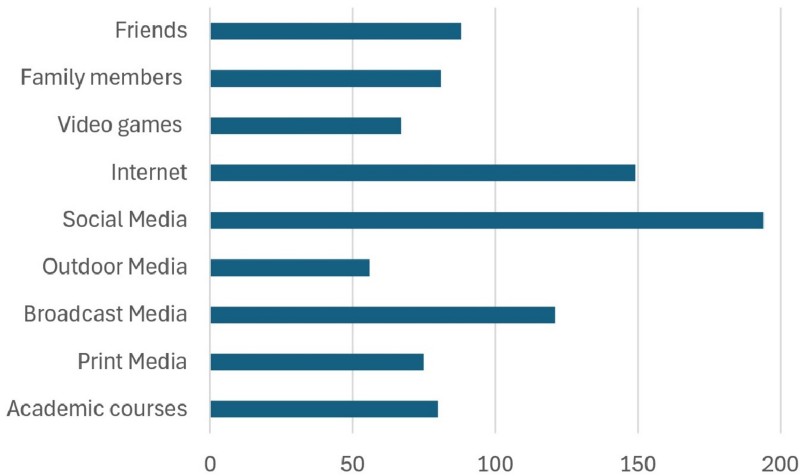

**Figure 1** Sources of information about AI among participants.

**Table 3 Chi-square analysis of attitudes against the categorical variables.**

| Demographic/ Continuous variable | Attitude (Categorical variable) | $\chi^2$ value | Degree of freedom | Statistical significance | Significance level |
|---|---|---|---|---|---|
| Age | Do you think that AI will play an important role in healthcare in the future? | 75.458 | 42 | 0.001 | Highly significant |
| | Do you believe that teaching in AI would be beneficial for their medical careers? | 77.658 | 42 | 0.001 | Highly significant |
| | Do you think that some specialties will be replaced by AI in the future? | 122.799 | 42 | 0.000 | Extremely significant |
| | Do you think confidentiality in the use of AI tools for primary healthcare is required? | 98.966 | 42 | 0.000 | Extremely significant |
| | Do you think whether AI will drastically change and revolutionize the medical field in 10 years? | 94.808 | 42 | 0.000 | Extremely significant |
| | Do you believe that all medical students should receive training in AI (mandatory need) as part of the curriculum to get their degree? | 95.943 | 42 | 0.000 | Extremely significant |
| Nationality | Do you think confidentiality in the use of AI tools for primary healthcare is required? | 7.553 | 2 | 0.023 | Significant |
| University | Do you think that AI will play an important role in healthcare in the future? | 51.870 | 32 | 0.015 | Significant |
| | Do you think that personal AI knowledge will improve the performance? | 87.171 | 32 | 0.000 | Extremely significant |
| | Do you believe that teaching in AI would be beneficial for their medical careers? | 64.998 | 32 | 0.000 | Extremely significant |
| | Do you think that some specialties will be replaced by AI in the future? | 91.192 | 32 | 0.000 | Extremely significant |
| | Do you think confidentiality in the use of AI tools for primary healthcare is required? | 101.405 | 32 | 0.000 | Extremely significant |
| | Do you believe that all medical students should receive training in AI (mandatory need) as part of the curriculum to get their degree? | 95.837 | 32 | 0.000 | Extremely significant |
| | Are you interested in improving your knowledge of AI? | 81.541 | 32 | 0.000 | Extremely significant |

(Continued)

| Demographic/ Continuous variable | Attitude (Categorical variable) | $\chi^2$ value | Degree of freedom | Statistical significance | Significance level |
|---|---|---|---|---|---|
| Program | Do you think that AI will play an important role in healthcare in the future? | 38.912 | 22 | 0.014 | Significant |
| | Do you believe that teaching in AI would be beneficial for their medical careers? | 106.055 | 22 | 0.000 | Extremely significant |
| | Do you think that some specialties will be replaced by AI in the future? | 34.498 | 22 | 0.044 | Significant |
| | Do you think confidentiality in the use of AI tools for primary healthcare is required? | 37.041 | 22 | 0.023 | Significant |
| | Do you think whether AI will drastically change and revolutionize the medical field in 10 years? | 40.864 | 22 | 0.009 | Significant |
| | Do you believe that all medical students should receive training in AI (mandatory need) as part of the curriculum to get their degree? | 66.728 | 22 | 0.000 | Extremely significant |
| | Are you interested in improving your knowledge of AI? | 49.784 | 22 | 0.001 | Highly significant |
| Year of study | Do you think that AI will play an important role in healthcare in the future? | 26.258 | 16 | 0.050 | Significant |
| | Do you think that personal AI knowledge will improve the performance? | 28.670 | 16 | 0.026 | Significant |
| | Do you believe that teaching in AI would be beneficial for their medical careers? | 71.391 | 16 | 0.000 | Extremely significant |
| | Do you think that some specialties will be replaced by AI in the future? | 51.823 | 16 | 0.000 | Extremely significant |
| | Do you think confidentiality in the use of AI tools for primary healthcare is required? | 36.983 | 16 | 0.002 | Significant |
| | Do you believe that all medical students should receive training in AI (mandatory need) as part of the curriculum to get their degree? | 42.533 | 16 | 0.000 | Extremely significant |
| GPA | Do you think whether AI will drastically change and revolutionize the medical field in 10 years? | 25.611 | 6 | 0.000 | Extremely significant |

the healthcare sector as significant. Notable associations were identified between this perception and factors including university classification, academic program, and year of study ($p < 0.05$).

The influence of personal familiarity with AI on performance was acknowledged by a considerable majority (82.3%) of the participants. Associations of significance were established between the level of personal AI knowledge and university classification as well as academic year of study ($p < 0.001$ for university classification; $p = 0.026$ for academic year).

A considerable portion (76.7%) of the respondents expressed the belief that integrating AI education into medical training would be advantageous for medical professionals. Significant relationships were identified among gender, age, university classification, academic program, and academic year of study ($p < 0.01$ for all factors).

A notable percentage (60.6%) of individuals speculated about the potential replacement of certain medical specialties by AI in the coming years. Significantly, correlations were identified with age, university classification, academic program, and academic year of study ($p < 0.05$ for all variables).

Concerns regarding confidentiality in the utilization of AI tools for primary healthcare were shared by 67.1% of the participants. Noteworthy connections were observed among gender, age, nationality, university classification, academic program, and academic year of study ($p < 0.05$ for all aspects).

A substantial majority (78.3%) of respondents concurred that AI is poised to bring about a transformative revolution in the medical field within the next decade. Significant associations were documented with university classification, academic program, and grade point average (GPA) ($p < 0.01$ for all variables).

Approximately half (51%) of the participants believed that incorporating AI training into medical curricula should be compulsory. This viewpoint was significantly linked to gender, university classification, academic program, and academic year of study ($p < 0.001$ for all variables).

An overwhelming majority (78.3%) of the respondents indicated a keen interest in enhancing their understanding of AI. This enthusiasm exhibited significant associations with gender, age, nationality, university classification, and academic program ($p < 0.05$ for all variables).

## Knowledge of AI

Table 4 presents the level of AI knowledge among students majoring in health-related fields. Approximately 43.8% of participants demonstrated an understanding of the fundamental principles of artificial intelligence. Statistically significant associations were observed between AI knowledge and gender, nationality, type of university, and academic program ($p < 0.01$ for all variables).

Various categories of AI based on capabilities were known by only 35.7% of the respondents. Significant relationships were identified with gender, age, type of university, program, and GPA ($p < 0.001$ for all).

Familiarity with the types of AI based on functions was 39.8%. Significant associations were found with sex, age, type of university, and programme ($p < 0.001$ for all).

A total of 49.4% of the participants were knowledgeable about AI programming-related cognitive abilities such as learning, reasoning, and self-correction. Gender, age, nationality, type of university, program, and GPA were significantly correlated ($p < 0.01$ for all).

The research emphasized a generally high level of awareness and positive outlook towards AI among students in the healthcare field, albeit with lower practical knowledge and participation in relevant courses. The connections observed between the demographic variables and responses indicate that factors such as gender, type of university, programme, and year of study impact students' awareness, attitudes, and behaviours concerning AI. While many participants acknowledge AI's potential influence on healthcare, fewer have delved deeply into the technical aspects or formal education on AI, pointing to a potential area for educational interventions. This underscores the necessity of integrating AI-related curriculum enhancements to better equip future healthcare professionals with a changing technological environment.

**Table 4 Chi-square analysis of knowledge against the categorical variables.**

| Demographic/ Continuous variable | Knowledge (Categorical variable) | $\chi^2$ value | Degree of freedom | Statistical significance | Significance level |
|---|---|---|---|---|---|
| Age | Do you have an understanding of the basic principles that underpin AI? | 84.794 | 42 | 0.000 | Extremely significant |
| | Did you know that the types of artificial intelligence based on capabilities are narrow (dedicated for one task), general (perform like human) and super AI (intelligent than human)? | 96.490 | 42 | 0.000 | Extremely significant |
| | Did you know that the types of artificial intelligence based on functions are reactive machines (Machine learning), limited memory (Deep learning), theory of mind (Emotions and interactions) self-awareness (future machines will have their own self- consciousness and sentiments)? | 106.381 | 42 | 0.000 | Extremely significant |
| | Did you know that AI programming cognitive skills: learning, reasoning and self-correction? | 78.697 | 42 | 0.001 | Highly significant |
| Nationality | Do you have an understanding of the basic principles that underpin AI? | 11.069 | 2 | 0.004 | Highly significant |
| University | Do you have an understanding of the basic principles that underpin AI? | 103.780 | 32 | 0.000 | Extremely significant |
| | Did you know that the types of artificial intelligence based on capabilities are Narrow (dedicated for one task), general (perform like human) and super AI (intelligent than human)? | 87.831 | 32 | 0.000 | Extremely significant |
| | Did you know that the types of artificial intelligence based on functions are reactive machines (Machine learning), limited memory (Deep learning), theory of mind (Emotions and interactions) self-awareness (future machines will have their own self- consciousness and sentiments)? | 94.608 | 32 | 0.000 | Extremely significant |
| | Did you know that AI programming cognitive skills: learning, reasoning and self-correction? | 101.701 | 32 | 0.000 | Extremely significant |
| Program | Do you have an understanding of the basic principles that underpin AI? | 61.718 | 22 | 0.000 | Extremely significant |
| | Did you know that the types of artificial intelligence based on capabilities are Narrow (dedicated for one task), general (perform like human) and super AI (intelligent than human)? | 52.833 | 22 | 0.000 | Extremely significant |
| | Did you know that the types of artificial intelligence based on functions are reactive machines (Machine learning), limited memory (Deep learning), theory of mind (Emotions and interactions) self-awareness (future machines will have their own self- consciousness and sentiments)? | 89.448 | 22 | 0.000 | Extremely significant |
| | Did you know that AI programming cognitive skills: learning, reasoning and self-correction? | 40.509 | 22 | 0.009 | Significant |
| Year of study | Did you know that the types of artificial intelligence based on capabilities are Narrow (dedicated for one task), General (perform like human) and Super AI (intelligent than human)? | 64.561 | 16 | 0.000 | Extremely significant |
| GPA | Did you know that the types of artificial intelligence based on capabilities are Narrow (dedicated for one task), general (perform like human) and super AI (intelligent than human)? | 25.686 | 6 | 0.000 | Extremely significant |
| | Did you know that the types of artificial intelligence based on functions are reactive machines (Machine learning), limited memory (Deep learning), theory of mind (Emotions and interactions) self-awareness (future machines will have their own self- consciousness and sentiments)? | 46.598 | 6 | 0.000 | Extremely significant |
| | Did you know that AI programming cognitive skills: learning, reasoning and self-correction? | 20.360 | 6 | 0.002 | Significant |

## DISCUSSION

This study offers significant insights into the awareness, attitudes, and knowledge of AI among health major students in public and private Saudi universities. The results underscore a generally positive perception of AI's role in healthcare but reveal notable gaps in practical knowledge and formal education on the subject.

### Demographic profile and influence on AI perceptions

The sample was predominantly composed of younger Saudi males from public universities, with a majority in the preclinical years. This demographic profile aligns with the trend of increasing technological receptivity among younger populations due to their native exposure to digital technologies from an early age (*Barham et al., 2018*). The associations observed between demographic variables- such as gender, type of university, and academic program- and AI-related variables highlight how these factors influence students' engagement with AI (*Pedro et al., 2019*). For example, gender differences in AI awareness could be a reflection of the varying educational focus and resources allocated to males and females in different fields, potentially necessitating targeted educational interventions (*Kim & Shin, 2021*). These findings echo international observations, such as those by *Veras et al. (2024)*, where demographic factors also shaped AI perceptions among health sciences students in other contexts (*Veras et al., 2024*).

### Awareness and knowledge gaps

The study revealed a high level of awareness among Saudi university students of artificial intelligence, indicating that Saudi students are curious and interested in technology and artificial intelligence. Most of the participants (96%) defined artificial intelligence correctly, which is explained by the recent widespread spread of artificial intelligence applications and its frequent mention in the media and educational discourse due to the large amounts of data it provides, as well as the improvement of computing capabilities and the high capabilities it has demonstrated in data processing, understanding patterns and relationships, accuracy of conclusions, and quality of decision-making in specific tasks (*Delello et al., 2023*). Despite all this awareness among the participating students, only 28.5% had attended courses related to artificial intelligence. This low attendance reveals a critical gap between theoretical awareness and practical application, indicating the difference between what individuals know and what they can actually apply in actual work situations. This gap may be the result of several factors, including a lack of practical training and the use of traditional educational methods that focus on theory without application (*Lee & Yoon, 2021*; *Pedro et al., 2019*). Bridging this gap is crucial to enhance productivity and efficiency, as individuals capable of applying knowledge effectively contribute more significantly to organizational and healthcare goals.

For the sources of information about artificial intelligence that were adopted by the students participating in the study, social media and the internet were the dominant channels for most students. It is true that social media platforms, such as Facebook, Twitter, Instagram, Snapchat and WhatsApp, allow information to spread much faster than traditional methods do and thus help to raise awareness quickly and effectively, but

they have many negative effects, such as distraction by giving rumours or biased or incorrect information (*Cope, Kalantzis & Searsmith, 2021*; *Li et al., 2024*). The use of these informal sources shows that we need to develop organized teaching plans and include them in schools. This will help make learning about artificial intelligence better and more comprehensive.

## Attitudes and expectations

Attitudes towards AI were largely positive. Most respondents (89.2%) believe AI will play a pivotal role in the future of healthcare, aligning with their high expectations for AI's contributions to clinical management (*Al Saad et al., 2022*). The relationships between this perception and the studied variables, such as the educational environment and the level of academic study, also indicate that students are aware of the role and importance of AI (*Sit et al., 2020*). This finding supports the need to improve curricula and enrich them with AI materials to keep pace with students' perceptions and ensure their readiness for the profession of the future.

The positive attitude towards AI continues, as the majority of participants (82.3%) acknowledged the impact of personal familiarity with AI on performance, reflecting Saudi students' awareness of the impact of personal experience on the quality of what a person can offer and how this person can exploit and adapt artificial education technologies in a beneficial way in the healthcare sector (*Ali Mohamad et al., 2023*). We found that this perception increases with increasing academic level and may differ from one university to another, which confirms the importance of education in shaping and formulating students' understanding of and competence in the use of AI technologies and programs, which will be positively reflected in public health later (*Ahmad et al., 2023*).

Advocating for AI education would be beneficial for medical professionals, as 76.7% of male and female students from all academic programs believe. This indicates a clear demand and desire to learn AI and train in its applications and programs (*Fazil et al., 2024*), as it will have a clear impact on increasing the level of health professions and effectively using technology to meet diverse needs (*Cheng et al., 2020*).

Despite all the previous strong positive trends, we found that only 60.6% of the participating students expected some medical specialties to be replaced by AI in the coming years, with some differences according to age, university classification, academic program, and year of study. Additionally, 67.1% of the participants were concerned about confidentiality in the use of AI tools in primary health care, with some differences between different demographic and academic factors. These results indicate students' concerns about their professional future and whether their roles will be replaced by programmed machines (*Li et al., 2022*). Hence, the importance and role of the educational institution in addressing these concerns through educational curricula will focus on the importance of building personal knowledge and ethical considerations and integrating them in the field of AI and how to benefit from its tools and applications (*Akgun & Greenhow, 2022*), ensuring that healthcare professionals are able to effectively manage privacy in their future careers (*Akgun & Greenhow, 2022*; *Pedro et al., 2019*).

Even with all the concerns that AI threatens students' professional future, 78.3% of them believe that AI is poised to transform the medical field better over the next decade, with little variation in this belief depending on the rankings of universities, academic programs, and students' GPA. This belief among students highlights their belief in the great potential of AI in healthcare (*Bohr & Memarzadeh, 2020*). This requires the development of customized educational strategies that prepare students for increasing technological advancements and enhance the integration of AI into the healthcare landscape (*Osunlaja et al., 2024*).

One surprising finding is that despite most students' preference for AI, half of them do not believe that they should receive AI training as part of their curriculum. This suggests a significant divide in perceptions regarding the importance of AI in their education. This may be explained by students feeling that they do not need AI immediately and that their current curriculum is sufficient to prepare them to be competent professionals (*Teng et al., 2022*). To address this divide, we reiterate the importance of integrating AI into the curriculum and better highlight the value of using AI tools and techniques and their potential impact on their future professional careers (*Chiu & Chai, 2020*).

Among the attitudes toward AI, 78.3% of the health science students who participated in the study expressed a strong interest in furthering their understanding of AI, which explains their strong desire to engage more deeply with this technology. This is explained by the fact that students are always motivated by curiosity, independence, and self-learning, and the use of AI encourages them to explore educational materials further and improve academic performance by helping them perform better on tests and exams through the provision of personalized reviews and learning materials (*Chen, Chen & Lin, 2020*; *Huang, Lu & Yang, 2023*). The study also indicated that this belief varied across demographic and educational backgrounds, reflecting differences in initial exposure to AI, the perceived importance of AI for specific medical specialties, or the influence of institutional attitudes toward the technology (*Stewart, Rybicki & Dwivedi, 2020*). These data emphasize the importance of everyone having equal opportunities to learn about AI and prepare to use these technologies effectively in their future careers (*Bohr & Memarzadeh, 2020*).

## Knowledge and technical competence

Despite strong awareness and positive attitudes, students' technical knowledge was moderate. Only 43.8% reported understanding AI's basic principles, with significant differences across demographic and academic factors. The result is logical given that AI is a new science worldwide and has only recently been included in educational curricula in recent years (*Stolpe & Hallström, 2024*). This moderate knowledge of the basics of this science can be attributed to differences in curricular focus or educational resources that affect the possibility of teaching AI in different universities and educational institutions (*Chen, Chen & Lin, 2020*; *Smit, Zoet & van Meerten, 2020*).

The study also highlighted AI types, which are classified into two categories: first, capabilities (narrow AI (dedicated to a single task), general AI (working like a human), and super-AI (smarter than humans)) and second, functions (interactive machines (machine

learning), limited memory (deep learning), theory of mind (emotions and interactions), and self-awareness (future AI with self-awareness)). When health college students were asked about both types of AI, only 35.7% of the participants were aware of capability-based AI typologies, and 39.8% of the participants were aware of function-based AI typologies. This lack of knowledge reveals a significant gap in students' understanding of the potential of diverse AI technologies, highlighting the need for more comprehensive AI education that effectively covers these fundamental distinctions (*Al Saad et al., 2022*; *Saghiri et al., 2022*). Significant associations with demographic and academic factors also suggest that students with stronger academic performance or specific educational experiences may have a better understanding, suggesting that educational backgrounds and demographic factors influence knowledge. This suggests that we need to educate students in general and health professionals in particular about how AI works, regardless of where they come from or their background (*Chung, Thaichon & Quach, 2022*; *Cope, Kalantzis & Searsmith, 2021*).

Regarding cognitive AI programming skills such as learning, reasoning, and self-correction, nearly half of the participants (49.4%) reported familiarity. The study further found that students with higher academic performance or those enrolled in technologically advanced universities demonstrated a better understanding of these skills. These findings highlight the importance of developing targeted educational interventions to enhance all students' comprehension of advanced AI concepts (*Cope, Kalantzis & Searsmith, 2021*; *Khanagar et al., 2021*; *Shin, 2020*).

After the knowledge of students is studied, a critical gap in AI education can be identified; therefore, it is necessary for universities and colleges to implement comprehensive training in the field of AI that covers not only the basic principles but also the capabilities and functions of more complex AI. This is important because we want to prepare them for jobs where they can use intelligent computer technology, such as assisting doctors, nurses, technicians in hospitals and clinics and many other jobs.

## Limitations

The current study was designed and performed to assess the awareness, knowledge, and attitudes of study participants regarding AI, with the intention of informing curriculum design and policy development for future learners. However, several limitations should be considered. The cross-sectional design limits the ability to establish causality and may have induced selection bias, potentially leading to the overrepresentation of certain demographics, such as younger males from public universities. Additionally, reliance on self-reported data could introduce response bias, as participants may overestimate their actual AI competence. The study also did not include follow-up assessments, limiting the evaluation of knowledge retention or skill development over time. Furthermore, complexity related to potential confounding variables may have influenced the observed associations. Future research should incorporate longitudinal or interventional designs—such as following a cohort through AI-integrated curricula—to better establish causal relationships and validate findings. Finally, further questions assessing AI literacy and the integration of cognitive skills need careful framing to ensure clarity. A pilot study is

recommended, which could involve either testing the consistency and validity of the refined questionnaire or evaluating the feasibility and impact of educational interventions.

## CONCLUSION

This study provides valuable insights into the awareness, attitudes, and knowledge of artificial intelligence among healthcare students in Saudi universities. While participants demonstrated high levels of awareness and generally positive attitudes toward AI, their overall technical knowledge was moderate, and significant gaps were observed in understanding practical AI tools and applications. This disconnect between theoretical knowledge and hands-on skills may be linked to reliance on informal learning sources, such as social media, which, while raising awareness, do not provide structured, practical training.

These findings carry important implications beyond Saudi universities: they highlight the urgent need for structured AI education in healthcare curricula globally, emphasizing practical, application-oriented learning alongside theoretical instruction. Educational institutions should prioritize integrating AI-related content into health programs to ensure that future healthcare professionals are equipped to meet the demands of an increasingly technology-driven landscape. Policymakers can use these insights to guide curriculum reforms, develop targeted training strategies, and promote equitable access to AI education across diverse student populations, thereby enhancing workforce readiness in the era of digital health transformation.

## ACKNOWLEDGEMENTS

The authors sincerely thank Prof. Mohamed M. Abdel-Daim, Professor of Pharmacology at Suez Canal University, Egypt, for his valuable efforts and contributions, which have been a great source of inspiration for this work. The authors confirm that Quillbot was used only for paraphrasing and proofreading purposes.

### Funding

The authors received no funding for this work.

### Competing Interests

The authors declare that they have no competing interests.

### Author Contributions

- Mai Albaik conceived and designed the experiments, performed the experiments, analyzed the data, prepared figures and/or tables, authored or reviewed drafts of the article, and approved the final draft.
- Saeed A. Al-Qahtani analyzed the data, authored or reviewed drafts of the article, and approved the final draft.

- Mohammad Jaffar Sadiq Mantargi conceived and designed the experiments, performed the experiments, analyzed the data, prepared figures and/or tables, and approved the final draft.
- Adel Alghamdi analyzed the data, authored or reviewed drafts of the article, and approved the final draft.
- Ikhlas A. Sindi analyzed the data, authored or reviewed drafts of the article, and approved the final draft.
- Ryan A. Sheikh analyzed the data, authored or reviewed drafts of the article, and approved the final draft.
- Mohamed Kamel analyzed the data, authored or reviewed drafts of the article, and approved the final draft.
- Lina A. F. Kurdi analyzed the data, authored or reviewed drafts of the article, and approved the final draft.

### Ethics

The following information was supplied relating to ethical approvals (*i.e.*, approving body and any reference numbers):

Ethical approval for this study was obtained from the Research Unit at Batterjee Medical College, Jeddah, Saudi Arabia (Reference No. RES-2023-0055).

### Data Availability

The data is available in the Supplemental Files.

### Supplemental Information

Supplemental information for this article can be found online at http://dx.doi.org/10.7717/peerj-cs.3255#supplemental-information.

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
