# Peer review of "Exploring perspectives on artificial intelligence: awareness, attitudes, and knowledge among health majors students at Saudi universities"

_PeerJ Computer Science, doi:10.7717/peerj-cs.3255_

## Round 0.1 · original submission · Major Revisions

·

Basic reporting

1. The abstract from lines 40 to 61 should be consolidated into a single paragraph for concise summary of the research.
2. Please ensure that all raw data, code and metadata used in the study are submitted in accordance with the data-sharing policy.
3. The language in the conclusion for lines 404 to 410 need revision as the current phrasing does not provide clarity and comprehension.

Experimental design

1. Line 49 mentions the use of descriptive statistics and chi-square analyses for the survey data. It is necessary to provide a justification for why these methods were selected over other statistical approaches for this analysis.
2. In the Subjects and Methods section, a detailed paragraph should be included outlining the process for conducting descriptive statistics and chi-square analyses. Need to specify the variables and characteristics analyzed in the statistical approaches.

Validity of the findings

1. The reference to enhancing the curriculum in Line 407 of conclusion section needs further clarification regarding the specific improvements that should be made.

Additional comments

1. The authors have presented comprehensive information in the results section.
2. Tables and figure attached are detailed enough for understanding the analysis.

·

Basic reporting

1. To enhance clarity and coherence, the article could more explicitly articulate the following:
• Problem Definition & Significance: Clearly outlining the problem the study aims to address and why it is important will help establish a stronger foundation for the research.
• Literature Gap & Research Contribution: Providing a more structured discussion on where the gap exists in the existing literature and how this study addresses it will further strengthen its contribution.
• Flow & Coherence: In some instances, improving the logical flow could enhance readability. For example, in lines 90–91, the statement on AI proliferation being limited is presented as a key motivation for the study. However, connecting this point more explicitly to the study’s objective and supporting it with relevant literature would improve clarity.
• Interpretation of Results: Rather than solely presenting the results, offering clearer explanations of their implications would make them more accessible to the reader. For instance, lines 187–189 could be expanded to help readers interpret the significance of the findings more easily.
2. While the background context is provided, the integration of literature directly relevant to the study could be strengthened. For example, several studies in the healthcare and education sectors focus on similar aspects, including systematic and meta-analytic reviews. Engaging with these sources would help clarify the research direction and critically identify the gap in the literature. This, in turn, would allow the study to extend existing work and make a more meaningful contribution to the field.
3. The article structure and the provided figure are appropriate. However, tables could be used more effectively to present descriptive and correlation results in a clear and succinct manner, reducing lengthy text while maintaining readability. Additionally, the raw data could not be located.
4. The results section would benefit from clearer interpretation. For instance, in lines 191–193, the statement: “Gender, type of university, programme, and study year were identified as having significant relationships with course attendance (p<.001 for all)” could be expanded to help readers understand the meaning and implications of these findings.
5. The survey design requires some refinement. Certain questions assume prior knowledge (e.g., Q22), contain double-barrelled elements (e.g., Q23), or may introduce response bias (e.g., Q21). Revising these aspects would strengthen the validity and reliability of the survey.

Experimental design

1. To develop original research, thoroughly explore existing literature to identify knowledge gaps. This will strengthen the foundation of your work and position your contribution within the field.
2. While you provided an objective, a clearly defined research question is missing. It is also critical to specifically address the critical aspect of the objective of this study "whether AI is viewed as innovation or threat." Additionally, include a dedicated section explaining how your research contributes to new knowledge in the field.
3. The investigation could be strengthened by: Adopting a more robust methodology; Designing a more effective survey instrument without bias; clearly articulating findings in relation to your research question
4. The methods section would benefit from: Providing a rationale for selecting the chi-square test when more powerful methodologies like ANOVA might be more appropriate to derive more meaningful comparison, especially with some useful demographic data that has been collected; including tests for non-violation of statistical assumptions; detailing the sampling methodology used in this study; explaining how variables were measured, including coding schemes and scale information (e.g. table-1 – how is it coded)

Validity of the findings

1. There are quite a few contemporary literatures in this area which was not explored. A thorough review of existing research would strengthen your theoretical foundation and methodology. The current flaws in survey design limit meaningful replication of your study.
2. No data was provided in your report, although the survey instrument was available. Including the actual data collected would allow for better evaluation of your findings and methodology.
3. While some conclusions were presented, they only covered part of the stated objective. The critical component examining "whether AI is viewed as innovation or threat" was not adequately addressed in your conclusions.
4. Consider a more robust methodological approaches; improved survey design to enhance replicability; and a clear connection between data collection, analysis, and research objectives

Additional comments

This study presents an interesting topic with strong potential, and there are several areas where further refinement could enhance its impact:
• Literature Review & Research Gap: Strengthening the engagement with relevant literature would help to better position the study. Clearly articulating the research gap and how this paper addresses it will add clarity to its contribution. See below some relevant literature

Mousavi Baigi, S. F., Sarbaz, M., Ghaddaripouri, K., Ghaddaripouri, M., Mousavi, A. S., & Kimiafar, K. (2023). Attitudes, knowledge, and skills towards artificial intelligence among healthcare students: A systematic review. Health science reports, 6(3), e1138.
• Al-Qerem, W., Eberhardt, J., Jarab, A., Al Bawab, A. Q., Hammad, A., Alasmari, F., ... & Al-Beool, S. (2023). Exploring knowledge, attitudes, and practices towards artificial intelligence among health professions’ students in Jordan. BMC Medical Informatics and Decision Making, 23(1), 288.
• Al-Dmour, R., Al-Dmour, H., Basheer Amin, E., & Al-Dmour, A. (2025). Impact of AI and big data analytics on healthcare outcomes: An empirical study in Jordanian healthcare institutions. Digital Health, 11, 20552076241311051.
• Serbaya, S. H., Khan, A. A., Surbaya, S. H., & Alzahrani, S. M. (2024). Knowledge, attitude and practice toward artificial intelligence among healthcare workers in private polyclinics in jeddah, Saudi Arabia. Advances in medical education and practice, 269-280.
• Mehta, N., Harish, V., Bilimoria, K., Morgado, F., Ginsburg, S., Law, M., & Das, S. (2021). Knowledge of and attitudes on artificial intelligence in healthcare: a provincial survey study of medical students. Medrxiv, 2021-01.
• Research Question: A more precise research question, explicitly framed in relation to gaps in the existing literature, will help sharpen the study’s focus and significance.
• Methodology & Data Collection: The methodological approach could be strengthened by considering more robust techniques where appropriate. Presenting results succinctly and ensuring that all underlying assumptions are tested and reported will enhance the study’s rigor. Additionally, refining the survey design—avoiding biases, double-barreled questions, and assumptions about respondent knowledge—will improve data reliability.
• Results & Interpretation: Providing a clearer interpretation of findings beyond statistical significance will strengthen the discussion. Elaborating on what the results imply in a broader context would add value.
• The "So What?" Question: Expanding on the study’s implications, limitations, and contributions would help clarify its relevance. Suggestions for future research directions will also be beneficial in positioning the study within the wider academic conversation.

---

## Round 0.2 · Major Revisions

The reviewer has provided valuable comments. Please read the details carefully and revise the work accordingly.

·

Basic reporting

It is commendable to see that the study reports ethical approval and makes raw data and SPSS files available. This level of transparency strengthens the credibility of the research, enhances reproducibility, and demonstrates adherence to good research practices. Such openness is particularly valuable in emerging fields, such as AI in healthcare education, where robust evidence and replicability are essential for advancing curriculum development and informed policy decisions.

Introduction

Consider condensing the introduction to keep it focused on framing the problem and motivating your study.

The introduction contains a comprehensive range of information, but the sequence of ideas feels somewhat fragmented. The text shifts between AI applications, benefits, risks, education, ethics, and Saudi-specific context in a manner that makes it difficult for the reader to follow a straightforward narrative. A more streamlined structure might strengthen the argument. For example, beginning with a global overview of AI in healthcare (applications, benefits, risks), then moving to the need for AI education (international and Saudi context), followed by highlighting specific research gaps, and concluding with a clear statement of the study aim. This approach would provide a smoother progression and allow the rationale for the study to emerge more naturally.

The introduction would benefit from referencing very recent empirical work on AI and health sciences education, such as the mixed-methods crossover randomized controlled trial by Veras et al. (2024) published in Digital Health. This study examined the experiences and perceptions of usability among health sciences students regarding ChatGPT-3.5, reporting both benefits (productivity, brainstorming, creativity, and supplementary learning) and concerns (accuracy, reliability, academic integrity, and the need for clear policies and guidelines). Integrating this evidence would make the introduction more current and align it with emerging discussions on how generative AI tools are perceived and adopted by students. It would also provide a strong rationale for your study’s focus on awareness, attitudes, and knowledge of AI among Saudi healthcare students, since it underscores the broader global trend of mixed opportunities and challenges. (1.
Veras M, Dyer J-O, Shannon H, et al. A mixed methods crossover randomized controlled trial exploring the experiences, perceptions, and usability of artificial intelligence (ChatGPT) in health sciences education. DIGITAL HEALTH. 2024;10. doi:10.1177/20552076241298485

Experimental design

The Subjects and Methods section provides useful detail on study design, sampling, and questionnaire development; however, it would benefit from closer alignment with established reporting guidelines such as STROBE (for cross-sectional studies) and SURGE (for survey-based research).
Inclusion of a completed STROBE/SURGE checklist as supplementary material is recommended, as this would ensure methodological completeness and facilitate peer evaluation.

Could the authors provide more detail on the pilot test outcomes (e.g., number of students involved, average time to completion, revisions made based on feedback)?

Validity of the findings

Results

The Results section is comprehensive, with clear presentation of demographic characteristics, awareness, attitudes, and sources of information regarding AI.

Some p-values are reported as “p=.028” while others are “p<.001.” Consistency in reporting format is recommended (APA or journal-specific style).

Figure 1 (sources of AI information) is referenced, but its description in text is overly detailed

Throughout the manuscript, the terms male and female are used when describing participants, but these are biological categories that refer to sex, not gender. The appropriate distinction is that:
• Sex (male/female) refers to biological differences.
• Gender (man/woman, boy/girl, non-binary, etc.) refers to socially constructed roles and identities.
If the study only collected data on biological sex, then the manuscript should consistently use the term sex and report categories as male and female. If gender identity was assessed, then the term gender (man, woman, other identities) should be used appropriately.
Table 2 - The table provides a summary of chi-square analyses but needs clearer structure, correct terminology, and complete statistical reporting (perhaps you can add df, effect size, consistent p-value format). Breaking it into smaller, more focused tables would improve readability. There is too much information in table 2.

Would the authors consider splitting the table into separate sections for Awareness, Attitudes, and Knowledge to improve readability?
Given the high number of significant associations, could the authors clarify which findings they consider most meaningful in practical or educational terms, beyond statistical significance?

Discussion

The discussion presents a wide range of findings, including awareness, attitudes, perceived risks, training gaps, and demographic influences; yet, the narrative sometimes feels fragmented due to frequent shifts between themes without clear transitions. Grouping results into broader subthemes (e.g., awareness and knowledge gaps, attitudes and expectations, concerns and ethical considerations, educational needs and recommendations) would improve clarity and reader engagement. Additionally, while the discussion cites relevant literature, more critical comparison with recent international studies (e.g., Veras et al., 2024, on health sciences students’ experiences with generative AI) could better situate the Saudi findings within a global context. Finally, the section could benefit from a stronger concluding synthesis that highlights the key implications. Although Saudi students demonstrate enthusiasm and curiosity about AI, there is an apparent mismatch between awareness and structured educational opportunities, underscoring the urgent need for formal AI integration into healthcare curricula.
Conclusion

Conclusion
1. The conclusion restates findings clearly, but could you strengthen the “so what” factor? For example, what do these findings imply for healthcare policy or curriculum development beyond Saudi universities?
2. You mention a “disconnect between theoretical knowledge and hands-on skills.” Could this be linked to reliance on informal learning sources (e.g., social media) as identified earlier? Clarifying this connection may reinforce the argument.
3. Would it be useful to briefly highlight the urgency of AI education in healthcare, given the rapid pace of technological integration?
Limitations
4. The limitations note the cross-sectional design and selection bias. Could you expand on how these factors may have specifically influenced your results (e.g., overrepresentation of certain demographics)?
5. Response bias is not mentioned — did you consider that self-reported awareness/knowledge might overestimate actual competence?
6. You recommend a pilot study; could you clarify whether this refers to piloting a refined questionnaire or piloting an educational intervention?
7. How might longitudinal or interventional studies (e.g., following a cohort through AI-integrated curricula) better address causality and validate your findings?

---

## Round 0.3 · accepted · Accept

Thanks to the authors for their efforts to improve the work. According to the comments of the reviewer, I believe this version is ready for acceptance. Congrats!

·

Basic reporting

The authors have addressed many of the comments and recommendations, and where applicable, have provided direct responses to my questions. They also included justifications in cases where changes were not made.

Experimental design

The authors have addressed many of the comments and recommendations, and where applicable, have provided direct responses to my questions. They also included justifications in cases where changes were not made.

Validity of the findings

The authors have addressed many of the comments and recommendations, and where applicable, have provided direct responses to my questions. They also included justifications in cases where changes were not made.